# Perspective Developing Successful Collaborative Research Partnerships with AI/AN Communities

**DOI:** 10.3390/ijerph18179089

**Published:** 2021-08-28

**Authors:** Jonathan Credo, Jani C. Ingram

**Affiliations:** 1College of Medicine, University of Arizona, Tucson, AZ 85721, USA; 2Department of Chemistry & Biochemistry, Northern Arizona University, Flagstaff, AZ 86011, USA; Jani.Ingram@nau.edu

**Keywords:** cultural competence and cultural safety, Indigenous data governance and data sovereignty, community development

## Abstract

In the United States, American Indian and Alaska Native (AI/AN) people are frequently under- or misrepresented in research and health statistics. A principal reason for this disparity is the lack of collaborative partnerships between researchers and tribes. There are hesitations from both academic Western scientists and tribal communities to establish new partnerships due to differences in cultural and scientific understanding, from data ownership and privacy to dissemination and project expansion. An infamous example is the mishandling of samples collected from the Havasupai Tribe by Arizona State University (ASU) scientists, leading to a legal battle between the tribe and ASU and ending in a moratorium of research with the Havasupai people. This paper will explore three successful and positive collaborations with a large and small tribe, including how the partnerships were established and the outcomes of the collaboration. In addition, the paper will provide perspective of what needs to be addressed by Western scientists if productive collaborations with tribal groups are to be established.

## 1. Introduction

In March 2021, the United States National Institutes of Health (NIH) announced the next step of their *All of Us* Research Program would “respectfully engage American Indian and Alaska Native (AI/AN) people” [1]. This announcement reflects the sentiment of many U.S. federal and state entities, as well as universities and non-profit organizations, seeking to work with AI/AN populations to address the under-representation of this group in research and healthcare. Many tribes and researchers have hesitations of working with each other due to several reasons, including a lack of trustworthiness of researchers by communities and a lack of cultural understanding as well as a belief that working with tribal groups is inherently difficult and time consuming by researchers [2,3,4]. To increase partnerships with tribal groups, NIH and many public universities have established training modules to educate academic researchers on how to conduct culturally appropriate research with vulnerable populations, including tribal people [3,4,5,6,7,8]. These modules are often informed from the work of researchers that have established successful research collaborations with Native American tribes that greatly benefit all parties involved and provide insights into the problems plaguing modern Native American communities [8]. Unfortunately, despite these focused programs, funding availability for working with tribal groups, and the success stories of other researchers, the disparity in AI/AN representation in healthcare and research is still prominent [9,10]. 

While understanding the nuanced history of a tribe may benefit a researcher interested in working with a specific Native American community, core differences in tribal and Western scientific cultures creates a division that stymies research collaborations. To provide perspective of why tribes are wary of scientists, this paper will summarize the Havasupai blood case. This paper will also detail the approach taken to developing a successful collaboration with two Native American tribes, one small and one large, and reflect on the lessons learned from these partnerships. If the divergence between Western science and AI/AN people is to be rectified, it necessitates Western scientists to develop innovative approaches that allow for research to be conducted in a culturally respectful manner, while still addressing the underlying research question. 

### Case Study: Havasupai Blood Case

In 1989, Dr. John Martin, an anthropologist at Arizona State University (ASU), was approached by members of the Havasupai tribe, a tribe located within the Grand Canyon and 1 of 21 federally recognized tribes in Arizona. The tribe hoped Dr. Martin could provide insight on why diabetes was increasing in their community and, if possible, help combat the chronic disease. As there had been other genetic links to diabetes in a different tribe, Dr. Martin enlisted the help of Dr. Therese Markow, a geneticist at ASU whose research involved genetic causes of disease. From 1990 to 1994, samples of blood and medical records were collected from approximately 400 members of the Havasupai Tribe, all of whom signed a broadly worded consent form that allowed the researchers to “study the causes of behavioral/medical disorders” [11,12]. The Havasupai members who consented in the study believed their samples would solely be used for the purpose of diabetes research and would help their tribe fight the disease. The ASU team would discover the previous genetic link to diabetes was not present in the Havasupai. However, research utilizing the Havasupai samples continued in other pursuits, including studies on tribal migration and origination, mental health, and alcoholism, all conducted without the Havasupai’s knowledge. While attending a dissertation presentation in 2003, Carletta Tilousi, a member of the Havasupai Tribe, learned her sample and those of her tribe had been used in studies that she viewed were never consented, including some studies centered on controversial and taboo topics in the Havasupai culture. In 2004, the Havasupai Tribe filed a case against the Arizona Board of Regents and Dr. Markow about the misuse of the samples [11,12]. The case would be settled out of court in 2010 with the tribal members receiving USD 700,000 in direct compensation, funds for a tribal clinic and school, and the return of the collected samples [11,12]. As a result of the Havasupai case, the Havasupai tribe passed a “Banishment Order” that barred all ASU researchers and employees from the Havasupai lands and stopped all ongoing research with the tribe. In addition, the case exemplified the concerns other Native American tribes had of working with outside researchers. To this day, many tribes are wary of entering research partnerships with outside entities and many continue to refuse to participate in genetic research studies. The effect in the scientific community has not been as widespread. While the Havasupai case serves as an example of the importance of communication and how “informed consent” can be misused, many researchers and institutional review boards still are not aware of the significance of this case or have not internalized any general lessons about tribal ethical considerations or cultural sensitivity [11]. 

## 2. Background

Dr. Ingram is a member of the Diné (Navajo) people and has worked with the Navajo Nation since joining the faculty at Northern Arizona University (NAU) in 2002, where she continues to collaborate with the Navajo people as a regent’s professor. She has worked collaboratively with many communities on the Navajo Nation to quantify the legacy of uranium mining contamination. Jonathan Credo is a Diné (Navajo) student that joined Dr. Ingram’s laboratory in 2010 and has assisted on many research projects related to uranium mining on the Navajo Nation, with an emphasis on water quality and availability [13]. As an MD/PhD student, Credo’s focus is on environmental health and toxicology in minority and tribal communities. Through this research, we have forged productive research partnerships with other tribal nations across the United States. 

While there are many factors that play into developing a new collaboration, there are salient lessons that can be applied to a general understanding. We are trained as physical scientists that have collaborated on numerous projects with both tribal and non-tribal communities, professionals, and students. Though we have partnered with social scientists on previous projects, we are not trained in the methodologies of these fields. The purpose of this paper is to contrast our experiences of developing productive continuing collaborations with two federally recognized Native American tribes. It is our intention to demonstrate that productive collaborations between tribal communities and Western-trained physical scientists are possible, and discuss some barriers that have made these types of projects difficult for some of our non-tribal colleagues. 

## 3. Collaboration One: Large Tribe Research Project

### 3.1. Methods 

In 2018, community members contacted Dr. Ingram at NAU to consult on two projects. One sought assistance in determining if a proposed agricultural plot adjacent to a former mine site would be safe to use, and the second project sought scientific counsel on developing legislation to address contamination from mining and how it impacts residents on the reservation. Following both requests, we coordinated with both teams of community members to have a planning meeting that would determine if our laboratory would be the best fit and how we should proceed. For the agricultural project, once we agreed to assist and our involvement was approved by community liaisons, a site visit was organized to see the proposed agricultural plots and meet with the field researchers from the U.S. Environmental Protection Agency (EPA). The project was presented to the entire community during a monthly town-hall-style community meeting and approval by the community council was given to start work in summer 2018. Prior to every sampling, the community liaison was notified, and a community representative was present. Topsoil (top 15 cm) samples were collected from the proposed agricultural plots and abandoned uranium mine sites. Topsoil samples were collected into individual whirl-pak bags along every 5 m on alternating sides along a 100 m transect, and two 100 m transects run in an “X” pattern at each site. All samples were taken back to NAU for preparation and analysis. Topsoil samples were removed from whirl-pak bags, placed on individual paper plates, and allowed to air dry for 72–96 h. After desiccation, samples were homogenized by crushing with a rubber mallet, dry sieved to 2 mm grain size, and placed in new whirl-pak bags for storage. Approximately 500 mg of prepared top-soil samples were placed into 55 mL CEM MARS Xpress PTFE digestion vessels with 10 mL trace metal grade nitric acid (HNO_3_) and digested using a CEM MARS6 microwave digestor following EPA protocol 3051A [14]. Digested soil samples were filtered using WHATMAN 0.45 µm PVDF w/PP filters into 10 mL trace-metal clean centrifuge tubes and stored at 4 °C until analysis. Immediately prior to analysis, samples were diluted 100-fold and spiked with 1 µg/L ^101^Ru as an internal standard and analyzed on a Thermo X-Series II inductively coupled-plasma mass spectrometer.

### 3.2. Results

Topsoil samples were collected from four sites—two agricultural (AGR) and two abandoned uranium mine sites (AUM)—with nine samples being collected at each site for a total of 36 topsoil samples. Figure 1 shows the concentration of uranium in µg/g (ppm). The average concentration of uranium differed at each site: AGR 1 was 0.17 µg/g, AGR 2 was 0.09 µg/g, AUM 1 was 0.18, and AUM 2 was 0.26. AGR 1 was the closest agricultural site to an AUM and was located approximately one kilometer southeast. All sites were below the average for uranium in soil, indicating no contamination concerns for the community [15]. In addition to uranium, at the request of the community, a qualitative survey analysis scanned the soils for 65 additional elements. The survey analysis revealed manganese (Mn) and strontium (Sr) were above in-house soil standards and the U.S. natural soil averages, and an additional investigation was suggested [16,17]. 

### 3.3. Discussion

The NAU team met with the EPA Region 9 team and the community liaison over a series of meetings from May to July 2018. The purpose of these meetings was to provide updates on the study and discuss the best approach to disseminate the results to the community. The results from the agricultural soil study were presented to the community at a town-hall-style meeting in August 2018. The community were relieved to hear there was no elevation of uranium in any of the soils sampled and that this finding agreed with the EPA findings of no elevation of radiation or radionuclides. There were questions regarding the significance of the elevated Mn and Sr values detected from the survey analysis. The questions primarily revolved around if these values would present an immediate or serious health concern for the community. The EPA team explained what the values meant from a regulatory perspective and why the values were flagged as concerning, though not alarming. We expanded upon this discussion by providing information on the medical implications of the exposure, complemented by Credo’s medical background. Ultimately, the community decided that there was no immediate concern and said it could potentially be the focus of a future study. 

## 4. Collaboration Two: Large Tribe Policy Project

### 4.1. Methods

For the policy initiative project, in the initial planning meeting, Credo was identified as the scientific consultant representative of the Ingram laboratory, given his background of working with tribes and quantifying contaminant exposure to tribal communities, in addition to his training as a medical student. Before continuing to work with state representatives, although they were from the tribal community, an inquiry was made regarding his involvement and if it would represent a concern to either NAU or the tribe. Following approval from both NAU and the tribe associated, Credo provided insight as a researcher and explained the importance of water quality and federal regulations. Additionally, as an MD/PhD student, he shared his experience as a training physician-scientist working with tribal members during the first two years of medical school during rural rotations on tribal lands, including the importance of understanding local environmental contamination issues and how they play a role in tribal member health.

### 4.2. Results

Following the series of policy planning meetings, “H.B. 2481–TPT; distribution tribal college compact” was drafted and presented to the Arizona House of Representatives. Credo was asked to attend the committee hearing and testify as an expert witness in support of the bill. During his testimony, he was asked to comment on the economic feasibility of the bill and speak at length on the health impacts of contaminant exposure to humans and livestock. The bill passed and was sent to the Arizona Senate, where, unfortunately, it failed to make it out of committee. 

### 4.3. Discussion

The outcome of the policy initiative was shared with the community during a monthly meeting of the community’s governing board. Both the governing board and the state representative were pleased with the outcome and felt that, although the bill was not passed, it was significant in informing other lawmakers on the plight facing the community.

## 5. Collaboration Three: Small Tribe Research Project

### 5.1. Methods 

In 2018–2019, Credo contacted the tribal administrator at a Native American tribe that we had never collaborated with. In this first correspondence, he introduced himself, his research interests of working collaboratively with tribal communities, the contamination-related research he was conducting in the region and inquired if the tribe had any concerns about contamination or exposure in their community. Credo was informed that the tribe had concerns of exposure to their people but were wary of entering a new partnership with an outside interest. To assuage concern, he worked with the tribal administrator over a series of months, initially over email and then in-person at three separate meetings, to build trust by sharing how we approached working with the other tribes across the United States and explaining how the science is conducted. During these discussions, Credo introduced the tribe to our laboratory and two colleagues at NAU, explaining that our laboratories could provide expertise in analytical chemistry, ecology, and ecotoxicology. These series of meetings with the tribal administrator eventually were presented to the tribal council who asked additional questions prior to approval; they were especially concerned about data sharing, privacy, and ownership. Following approval, the tribal council established a working group comprised of six tribal members that would coordinate with the NAU scientists. The working group would serve as the go between NAU and the tribal council and would provide monthly updates. 

### 5.2. Results

After obtaining approval from the tribal council, we received pilot seed money from an internal NAU grant competition. The purpose of the grant was to pay for meetings with tribal communities amenable to collaborating with NAU researchers and foster a healthy and culturally appropriate partnership. Our team proposed a series of four meetings, with two at NAU and two at the tribal headquarters. The purpose of these meetings would be to identify contamination concerns of the community, the best approach to quantify the exposure, and how to address the exposure (e.g., policy initiatives, education, mitigation efforts, etc.). During the first meeting, held in fall 2019, Credo presented upon the ongoing research he was carrying out in the region with other communities including the potential health impacts from the environmental exposure. The tribal partners shared their concerns about the environmental contamination in the region and provided a tour of the reservation lands. All subsequent meetings were held virtually due to the COVID-19 pandemic, but the meetings were still fruitful. In these meetings, our teams identified the best approach to gather preliminary information on exposure from environmental contamination, how the data would be shared and disseminated, and deadlines for data collection to apply for continuing funding from governmental sources (e.g., the NIH, EPA, the National Science Foundation, etc.). 

### 5.3. Discussion 

Despite the COVID-19 travel restrictions necessitating virtual meetings, the collaborative partnership between NAU and the tribe has continued to grow. Arguably, the move to virtual meetings was beneficial and contributed greatly to forging a strong trusting partnership much faster than anticipated when working with smaller tribes. A significant factor for this trust building was more time allotted to topics of particular concern for the tribe, including data privacy and sharing, explanation of the scientific approaches, and health implications of contaminant exposure. Additionally, the virtual meetings facilitated creation of focus groups that would oversee different aspects of the proposed project (e.g., field work, grant writing and funding, dissemination, etc.). The combination of these benefits also instilled a level of familiarity among the members who were willing to express concerns as they arose, either over email or requesting short one-hour video conferences. The short video conference format also served to keep meetings brief and on topic, staved off the meeting fatigue typically experienced during tribal research collaborative planning meetings, and respected other time commitments of the partners. As travel restrictions eased and individuals became fully vaccinated, field work was conducted in Spring and Summer 2021. The field work collected samples for preliminary analysis that will be included as pilot data when applying for other funding.

## 6. Conclusions

For researchers and groups wanting to develop research partnerships with AI/AN populations, it is important to remember that trust building and project development takes time, and the comfort of the tribe is paramount. Researchers must remember that these populations may be wary of outsiders, as many communities could have a negative history or negative perception of outsider involvement. Even once trust has been gained and a project is underway, it is all too easy for that trust to be eroded if a researcher or group is not careful. As Western-trained scientists, we can appreciate the arguments our colleagues make about working with tribes and the integrity of a project. At the same time, being Native American, we can empathize with tribal communities and their concerns about Western scientists. This approach expands beyond our experiences, as our other Western-trained colleagues of tribal descent have shared similar experiences of understanding both sides of the argument and having to “navigate” the best course of action [18]. Borrowing from our own Navajo culture, where hair is an important cultural icon, Indigenous scholars can think of their cultural and scientific upbringing as a braid: weaving two separate identities together to make a stronger unit. In our experience, some common pitfalls Western scientists and tribes have difficulty agreeing on are data ownership and dissemination, transparency, and equity in the partnership.

For any tribe we have collaborated with, one of the first concerns the tribe brings up is data ownership and dissemination. In the United States, our scientific training tells us that the data belongs to the scientist or the affiliated institution because it was our expenditure of resources that transformed the data into something meaningful. Additionally, samples collected are at our discretion to keep or destroy because they may have latent usefulness (e.g., longitudinal studies, comparative studies, etc.). We believe we possess ownership over samples once an individual signs an informed consent, as it is the implicit belief that a signed consent form also signs away ownership [11]. Further, as part of owning the data, it is at our discretion on how to disseminate the data (e.g., manuscripts, grant application, presentations, etc.). However, in our experiences with working with tribes, this notion is the opposite: tribes want assurance that the data belongs to them, and they have oversight regarding what information is shared or disseminated. If researchers and groups wish to maintain a working relationship with tribes, they must accept that the data belongs to the tribe. This does not mean, however, that the data will never be used constructively. Rather, at the onset of starting any collaboration, there needs to be a discussion on data sharing. While this may seem obtuse, data ownership placed under the auspices of the tribe provides a level of protection to the tribe. As exemplified in the Havasupai case, tribes understand the dual nature of data: it can provide insights to address questions and concerns of the community, but it can also shed light on potential sensitive information that ultimately may harm the tribe. An additional fear is that once the samples or data have been collected, the tribe will not see any benefit from the researcher. This was also reflected in the Havasupai case, where tribe members felt dozens of scientists built their careers or found success at the expense of the Havasupai tribe that never benefited from the work [11,12]. Placing ownership in the hands of the tribal communities is already a widely accepted framework in Canada, and can provide an excellent teaching opportunity for Western-trained scientists in the United States seeking to collaborate with tribal entities [19,20]. 

Tribes, much like most people, appreciate transparency, and the best approach for starting a collaborative partnership is to be as upfront as possible with the tribe. It is important early in the collaboration for the scientific team to express their needs and wants for the partnership and give space for the tribe to do so as well. Depending on the size of the tribe, this can take shape in many ways. A large tribe may have additional resources to push the proposal along, but conversely, can also have additional legislative bodies that need to agree on the work (e.g., independent institutional review boards, departments that oversee an aspect of the work, etc.). While a small tribe can seem more convenient, the lack of resources and expertise in understanding collaborations between tribal entities and outside entities can also represent an issue. At all times, scientists need to work as advocates on behalf of the tribe and put the interests of the community ahead of their own. Regardless of the size of the tribe, oversight and reporting is a constant commitment in these situations. Proposing regular update meetings to provide significant feedback or results as well as seeking permission when transitioning to a new step should not come as a surprise and can build the foundation for a trusting collaboration. An example from our own experiences included a colleague somewhat familiar with tribal collaborations that wanted to ignore the wishes of the tribe. Their rationale was that the science would be stronger if we proceeded in a different manner to what was proposed to the tribe and what the tribe expressed and agreed upon. They reasoned the tribe would be fine as they would not learn of the breach of trust and would be pleased with the result. As with any relationship, indiscretion in a partnership, regardless of the justification, will undermine any progress made. The rules and boundaries established at the onset should always be respected.

Scientists should enter a partnership with a tribe with the acceptance that it is truly a partnership, with neither party *needing* the other and both parties being equals. As part of that equity, scientists need to understand and respect the idea that tribal collaborations take time and resources to develop just as it takes time to conduct science. As evident with our own examples, regardless of the situation, there were various meetings with the community to understand the scope of the work as well as becoming acquainted with the team throughout the entire process. To facilitate this understanding, it is important to remember that AI/AN tribes are organized sovereign governments that have their own rules and regulations. Additionally, scientists need to accept that tribes possess their own expertise that may benefit the project. In the past decade, this notion has become more commonplace under the name of Traditional Ecological Knowledge (TEK) or Traditional Knowledge and has been recognized by the NIH, especially in the fields of environmental health and biomedical sciences [21]. TEK can broadly be described as Indigenous knowledge preserved in a tribe’s culture that provides a foundation to teach younger generations how to interact with their environment [22]. It is important to note that TEK is specific to each geographic location and application. For our interactions with tribal communities, it has proven a successful and useful approach for environmental analytical chemistry and exposure science. The previous paternalistic notion that the Western scientist or physician knows best will often hinder or harm the development of a collaboration. While tribes may not be trained in the same techniques or terminologies as a Western scientist, they know better than anyone else the land they inhabit or the problems they live with daily. The TEK framework provides an excellent opportunity for a two-way exchange of knowledge that can deepen the trust and likelihood for a successful collaboration. In fact, utilizing TEK can save time or aid in ensuring the success of a project [23,24]. Further, utilization of TEK is an easy approach to demonstrate your team values the equity of the relationship and the tribal communities’ knowledge base. In our own experience, a common difficulty with working with tribes is community involvement or securing support from the community at the onset of a project. In another example, from discussions with the tribe, our team learned which social media outlets the tribal members used frequently. Additionally, we learned that the tribe would not be invested into a project without public endorsement from the tribal council such as involvement of an elder or council member. 

Tribal collaborations can manifest in a variety of ways and can distinctly go against the training received as a Western scientist [25]. However, working with tribes does not represent an insurmountable goal. Instead, it necessitates a willingness on the part of the Western-trained scientist to change and adapt from dogmatic thinking. For a successful partnership, it is essential we view tribal partners as equal, value their contribution, and respect their sacrifice. Two-Eyed Seeing provides an example of how Western-trained Indigenous scholars have been able to reconcile the need to respect tribal sovereignty while still conducting *good science* [26,27]. Adopting these values should not come as a surprise, as many minorities and disenfranchised populations have a similar unease when working with scientists and physicians [28,29]. Lastly, much like there are regional differences in the United States within the dominant culture, one cannot treat all AI/AN communities the same. Each AI/AN population will have nuanced differences that require a commitment of time, resources, and energy to ensure a successful culturally appropriate partnership with tangible positive outcomes.

## Figures and Tables

**Figure 1 ijerph-18-09089-f001:**
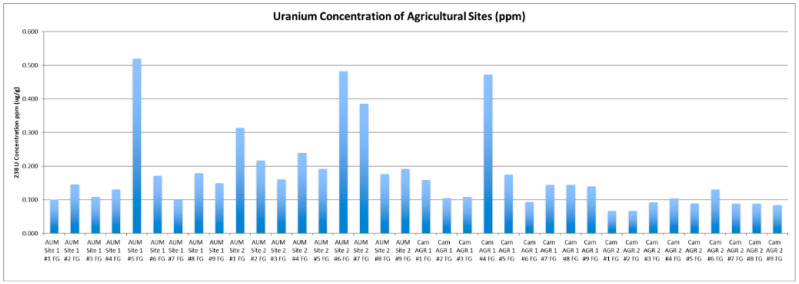
Uranium concentration in soil samples collected near proposed agricultural sites. Displayed are abandoned uranium mine (AUM) and agricultural sites (AGR).

## Data Availability

Not applicable.

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
