# Peer review of "Perspective Developing Successful Collaborative Research Partnerships with AI/AN Communities"

_ijerph, 2021, doi:10.3390/ijerph18179089_

Round 1
Reviewer 1 Report
Thank you for the opportunity to read this paper. Overall it is well written and the potential impact is high. To see the paper led by an Indigenous graduate student, mentored by an Indigenous researcher, is a strength of the paper. The ways in which the examples are written – the depth of relationship, etc – are also a strength. But it is my view that significantly more context can and should be provided about the current base of knowledge and method/ologies that this paper both grows from and contributes to. There is a healthy base of knowledge and critique in the published literature around community-based/ Indigenous/ ethical collaboration, and the failure of the authors to contextualise this work against that base of research is a major limitation. There are a number of other key issues that should be addressed to strengthen this paper & make it suitable for publication. These are identified below:
- There is absolutely no reference provided to the international readership or literatures on this topic. To me this is problematic as it evolves from the author’s data and experiences only, which is not how scholarship ought to be carried out (especially so for a international journal). The problem identified in this paper is global in nature and not specific to the narrow scope, literature and examples provided here. It is my view that the whole paper needs contextualising more broadly within the national and international examples and conversations around methods/ methodologies as they pertain to Indigenous health research and specifically to the ETHICS of doing Indigenous health research. OCAP is a principle that has been accepted and practiced in Canada for many years now – the idea that any scientist could “own” data collected with or for Indigenous communities is contested, and mainly viewed as unethical.
- The overall tone of this paper is authoritative, but incomplete. The writing is very much about what the authors (and really from the perspective of the 1st author!) have learned through the three presented cases. This is great, and deserving to be heard - but these messages must be contextualized against what is already known in the wider field (which is not new and indeed growing every day!). Please do the background literature necessary to adequately frame and discuss what you have learned, and what your findings contribute to.
- The way the cases are presented – each with their own distinct method/ results/ discussion – run a bit counter to the overall tone/ message of the overall paper. I see that these cases are used to provide how/ what the local cases looked like and how the collaboration evolved and was useful in determining collaborative decision making, but there seems to be a lot of time/ energy spent here that could otherwise be spent on literature/ methods/ologies to frame how collaborative env/health research has grow and what it means in the US for tribal communities, but also urban organizations and other Indigenous peoples and communities who share similar goals.
- That the first author’s voice becomes “I” in this paper makes me wonder how/where the 2nd author fits into this work. Our students/ trainees do not simply come to learn and be scholars – thus, it seems to me that the role on mentorship in building the next generation of Indigenous PhD’s is a central part of learning to be/do the important collaborative work shared here. Would the authors consider unpacking this a little in the overall discussion? The idea of knowledge transfer and capacity building is key to this work – both on the training front as well as in our collaborations.
- Traditional Ecological Knowledge is adopted late in the paper but never defined or supported from the literature. Where does this concept come from and how have others engaged with it as a means to support improved collaboration?
- Two-Eyed Seeing (Etuaptmumk in Mi'kmaw) embraces “learning to see from one eye with the strengths of Indigenous knowledges and ways of knowing, and from the other eye with the strengths of mainstream knowledges and ways of knowing, and to use both these eyes together, for the benefit of all,” as envisaged by Elder Dr Albert Marshall could be a very useful concept for this paper.
Author Response
1. There is absolutely no reference provided to the international readership or literatures on this topic. To me this is problematic as it evolves from the author’s data and experiences only, which is not how scholarship ought to be carried out (especially so for a international journal). The problem identified in this paper is global in nature and not specific to the narrow scope, literature and examples provided here. It is my view that the whole paper needs contextualising more broadly within the national and international examples and conversations around methods/ methodologies as they pertain to Indigenous health research and specifically to the ETHICS of doing Indigenous health research. OCAP is a principle that has been accepted and practiced in Canada for many years now – the idea that any scientist could “own” data collected with or for Indigenous communities is contested, and mainly viewed as unethical.
Response: We thank the reviewer for their comments and concern that the perspective does not provide an extensive international focus. We have provided some additional references in the “Conclusion” that provide an additional international focus (lines 264, 292-295, 355-359). Specifically, we have added references on OCAP, which we thank the reviewer for suggesting. However, the call for the special addition did not specifically state that articles need to be international in focus. As such, we felt that we could provide valuable information and discussion of our own experiences of working with AI/AN populations in the United States.
2. The overall tone of this paper is authoritative, but incomplete. The writing is very much about what the authors (and really from the perspective of the 1stauthor!) have learned through the three presented cases. This is great, and deserving to be heard - but these messages must be contextualized against what is already known in the wider field (which is not new and indeed growing every day!). Please do the background literature necessary to adequately frame and discuss what you have learned, and what your findings contribute to.
Response: The purpose of this manuscript was to provide a perspective of more than a decade of successful collaborative partnerships with tribal communities from the lens of two physical scientists. Neither author is trained as social scientists, and we have included language in the “Background” section of the manuscript stating this fact (line 98-106). As mentioned in our response to the first comment, we thought our experiences as physical scientists working with tribal communities and both tribal and non-tribal communities, professionals, and students for over a decade would be insightful especially in the context of what was requested by the special issue.
3. The way the cases are presented – each with their own distinct method/ results/ discussion – run a bit counter to the overall tone/ message of the overall paper. I see that these cases are used to provide how/ what the local cases looked like and how the collaboration evolved and was useful in determining collaborative decision making, but there seems to be a lot of time/ energy spent here that could otherwise be spent on literature/ methods/ologies to frame how collaborative env/health research has grow and what it means in the US for tribal communities, but also urban organizations and other Indigenous peoples and communities who share similar goals.
Response: We thought that it would be helpful to provide context for the reader to understand the environmental research project and the interactions with the community as we drafted this manuscript as a perspective, and not a research project or a commentary on how indigenous/tribal/etc. collaborative research has evolved. We have opted to keep the format in the manuscript as is.
4. That the first author’s voice becomes “I” in this paper makes me wonder how/where the 2ndauthor fits into this work. Our students/ trainees do not simply come to learn and be scholars – thus, it seems to me that the role on mentorship in building the next generation of Indigenous PhD’s is a central part of learning to be/do the important collaborative work shared here. Would the authors consider unpacking this a little in the overall discussion? The idea of knowledge transfer and capacity building is key to this work – both on the training front as well as in our collaborations.
Response: We have changed the language in the manuscript to reflect both author’s experiences and have removed the implication that these experiences reflect only that of one author. Additionally, we have provided more context in the “Background” section that introduces each author and why each other has insight to contribute. As well as introducing the more than ten-year collaborative partnership between both partners and how we have approached working with tribal communities.
5. Traditional Ecological Knowledge is adopted late in the paper but never defined or supported from the literature. Where does this concept come from and how have others engaged with it as a means to support improved collaboration?
Response: We have provided a citation in the manuscript that discusses how we, Dr. Jani Ingram especially, have used TEK successful in partnerships with the Navajo Nation. We have also provided a definition in the text (lines 328 – 333) that introduces TEK. Additionally, we mentioned specific for our fields (environmental analytical chemistry and exposure science) and in the communities we have worked with, TEK has been successful (lines 333-335).
6. Two-Eyed Seeing (Etuaptmumk in Mi'kmaw) embraces “learning to see from one eye with the strengths of Indigenous knowledges and ways of knowing, and from the other eye with the strengths of mainstream knowledges and ways of knowing, and to use both these eyes together, for the benefit of all,” as envisaged by Elder Dr Albert Marshall could be a very useful concept for this paper.
Response: We have included references to Two-Eyed Seeing in the closing paragraph of the “Conclusion” (lines 355-359). We have also included language at the start of the “Conclusions” section of a related phenomenon described by our indigenous colleagues as well as the citation that supports this observation of the phenomenon (lines 261-268).
Reviewer 2 Report
This perspective describes three successful collaborations between American Indian and Alaska Native (AI/AN) communities and researchers. Authors assert a need for their study due to the disparity in AI/AN representation in healthcare and research, the need to foster greater cultural understanding among researchers, and the benefits of successful research collaborations with AI/AN communities.
The feedback is as follows:
- Line 10 - Authors should consider specifying what type of researchers are being referenced. It may be assumed that the paper discusses academic researchers since a university is mentioned. It may be helpful to delineate that the paper focuses on academic researchers (e.g., versus government, private, or other researchers).
- Line 24 – the acronym NIH (National Institutes of Health) should be written out in full the first time it is introduced.
- Line 30 – With the mention of “lack of trust”, who is the lack of trust among? If among the AI/AN communities, it may be worth considering how to phrase this sentiment so that it does not put the onus on the community members. For instance, there could be a mention of the ‘lack of trustworthiness’ of researchers.
- Line 51-84 – the Case study: Havasupai Blood Case is a very compelling story and provides insightful contextual information into why AI/AN communities may be wary about entering community partnerships.
- Line 98 – Check punctuation and wording in “In 2018 community members contacted Dr. Ingram and me at NAU for assistance on 98 two projects”.
- Lines 98-101 – In “In 2018 community members contacted Dr. Ingram and me at NAU for assistance on two projects. One sought assistance in determining if a proposed agricultural plot adjacent to a former mine site would be safe to use, and the second project sought scientific council on developing legislation to address contamination from mining and how it impacts residents on the reservation.”, what prompted the request from community members? What this is a historical or contemporary issue? Where there any previous instances that prompted concern?
- Overall, this is a unique, insightful, and commendable effort. Attending to some clarifying questions may help to improve quality of the paper.
Author Response
This perspective describes three successful collaborations between American Indian and Alaska Native (AI/AN) communities and researchers. Authors assert a need for their study due to the disparity in AI/AN representation in healthcare and research, the need to foster greater cultural understanding among researchers, and the benefits of successful research collaborations with AI/AN communities.
The feedback is as follows:
1. Line 10 - Authors should consider specifying what type of researchers are being referenced. It may be assumed that the paper discusses academic researchers since a university is mentioned. It may be helpful to delineate that the paper focuses on academic researchers (e.g., versus government, private, or other researchers).
Response: We have specified that we are talking about academic researchers (lines 11 and 34).
2. Line 24 – the acronym NIH (National Institutes of Health) should be written out in full the first time it is introduced.
Response: We have spelled out the NIH acronym and specified it is a U.S. institution at its first appearance (line 24).
3. Line 30 – With the mention of “lack of trust”, who is the lack of trust among? If among the AI/AN communities, it may be worth considering how to phrase this sentiment so that it does not put the onus on the community members. For instance, there could be a mention of the ‘lack of trustworthiness’ of researchers.
Response: We have adopted the reviewer’s suggestion and specified a “lack of trustworthiness of researchers by communities” and a “lack of cultural understanding…by researchers” (lines 29-33)
4. Line 51-84 – the Case study: Havasupai Blood Case is a very compelling story and provides insightful contextual information into why AI/AN communities may be wary about entering community partnerships.
Response: We thank the reviewer’s thought and comment about the Havasupai Blood Case. For both authors it has also served as an example of the tenuous relationship between tribes and United States western scientists.
5. Line 98 – Check punctuation and wording in “In 2018 community members contacted Dr. Ingram and me at NAU for assistance on 98 two projects”.
Response: Changed the language to reflect the community reached out to Dr. Ingram as a consultant. The subsequent sentences build upon the narrative to explain how the other author (Jonathan Credo) was included in these partnerships.
6. Lines 98-101 – In “In 2018 community members contacted Dr. Ingram and me at NAU for assistance on two projects. One sought assistance in determining if a proposed agricultural plot adjacent to a former mine site would be safe to use, and the second project sought scientific council on developing legislation to address contamination from mining and how it impacts residents on the reservation.”, what prompted the request from community members? What this is a historical or contemporary issue? Where there any previous instances that prompted concern?
Response: We have expanded the “Background” section of the paper to discuss the more than ten-year collaboration between the authors. Additionally, the “Background” section builds upon how Dr. Ingram has focused her academic career to helping tribal communities, which we feel provides context why tribal communities near NAU reached out to her. We have opted to continue not identifying either of the tribes that we are mentioning in this manuscript out of respect for the tribe’s privacy.
7. Overall, this is a unique, insightful, and commendable effort. Attending to some clarifying questions may help to improve quality of the paper.
Response: We thank the reviewer for their productive comments and positive feedback. We hope that our experiences as physical scientists collaborating successfully with tribal communities for more than ten years will prove useful for the body of science.